# Variable crab camouflage patterns defeat search image formation

Jolyon Troscianko [1✉], Ossi Nokelainen [2], John Skelhorn [3] & Martin Stevens [1]

Understanding what maintains the broad spectrum of variation in animal phenotypes and how this influences survival is a key question in biology. Frequency dependent selection – where predators temporarily focus on one morph at the expense of others by forming a "search image" – can help explain this phenomenon. However, past work has never tested real prey colour patterns, and rarely considered the role of different types of camouflage. Using a novel citizen science computer experiment that presented crab "prey" to humans against natural backgrounds in specific sequences, we were able to test a range of key hypotheses concerning the interactions between predator learning, camouflage and morph. As predicted, switching between morphs did hinder detection, and this effect was most pronounced when crabs had "disruptive" markings that were more effective at destroying the body outline. To our knowledge, this is the first evidence for variability in natural colour patterns hindering search image formation in predators, and as such presents a mechanism that facilitates phenotypic diversity in nature.

[1] Centre for Ecology and Conservation, College of Life and Environmental Science, University of Exeter, TR10 9FE Penryn, UK. [2] Department of Biological and Environmental Science, University of Jyväskylä, Jyväskylä, Finland. [3] Biosciences Institute, Faculty of Medical Sciences, Newcastle University, NE2 4HH Newcastle upon Tyne, UK. ✉email: jt@jolyon.co.uk

Across nature there exists enormous variation in the colour patterns of animals, plants, and other organisms[1–4], and understanding the drivers of this is central to answering many core questions in biology. Camouflage is a powerful and widespread means of avoiding detection or recognition, and can be achieved via a variety of strategies. Moreover, camouflage patterns often show high levels of intraspecific variation even within the same locality[2,3,5]. One of the most widely suggested but seldom directly-tested causes of this diversity, is for defence against predator cognition and search behaviour[6–8]. Poulton[9] first noted that searching for one prey type at a time is easier than looking for several types, and Luuk Tinbergen[10] suggested that predators experience a perceptual change in their ability to detect prey types that are encountered repeatedly; i.e. they form a search image for those prey. This enables predators to search more effectively for common prey types[11], and capture these disproportionately often compared to prey with rarer phenotypes. One outcome is thought to be negative frequency-dependent (or 'apostatic') selection[6,7,12] which can, over time, lead to fluctuations in the frequency of different phenotypes.

Classic experiments by Pietrewicz and Kamil[6], in which blue jays (*Cyanocitta cristata*) searched projection slides for camouflaged *Catocala* moths, showed that when jays saw runs of the same moth species, their performance improved over time. By contrast, when jays were presented with a mixture of moth species, there was little improvement in performance, presumably because search image formation was inhibited. Further experiments using computer-generated prey have supported the idea that that jays form search images for common prey types, and have demonstrated that when prey patterns are allowed to "evolve" based on a genetic algorithm that gives undetected individuals a higher likelihood of reproducing, the phenotypic diversity of the concealed prey increases. Moreover, this diversity is in the form of continuous variation rather than discrete morphs[13,14]. This suggests that high intraspecific diversity may impair predator search efficacy and in turn bring survival benefits to a variable prey.

There is also evidence that variation in camouflage strategy could be favoured by selection. Camouflage can be achieved through a range of different mechanisms. For example, prey can use background matching, where they resemble the colour and pattern of the general environment, or disruptive coloration, where relatively high contrast markings break up the body outline[15,16]. And in natural populations, phenotypic variation can take the form of continuous variation in a single camouflage strategy, discrete variation in the camouflage strategy used, or a combination of the two. Experiments in which human observers search for camouflaged targets have demonstrated that both the camouflage strategy used, and specific features of camouflaged targets (e.g. internal pattern contrast), can interfere differently with various aspects of predator learning or attention[17,18]. Disruptive camouflage is particularly effective at impairing search image formation, and this effect is enhanced when observers have been searching for more easily-detected prey immediately before encountering runs of disruptive prey[19]. However, this work focuses exclusively on understanding how search images are formed for broad camouflage strategies when prey individuals are highly variable (i.e. when all individuals with the same camouflage strategy type have a different phenotype). In addition, these studies, and those more widely on search image formation, have never used natural intraspecific variation in prey phenotypes. Thus, whilst it is theoretically possible for both continuous variation in camouflage patterns and variation in camouflage strategies to inhibit search image formation, there is as yet no direct evidence that the intraspecific variation observed in the patterns of natural prey has this effect.

Green shore crabs (*Carcinus maenas*) offer an excellent system for testing the relationship between camouflage, phenotypic variation, and search image formation. They appear to show a number of morphs, with considerable variation in appearance around the key elements of each morph type (Fig. 1a)[20,21]. In addition, there exists extremely high levels of intraspecific variation in colour and pattern within the same locations and habitats[22,23], meaning that the system allows us to investigate both continuous and discrete (i.e. morph-based) effects on search image formation[21]. The natural predators of green shore crabs are highly visually guided, including birds such as corvids, gulls and shorebirds (all plausibly tetrachromats), and numerous species of fish (such as gobies, blennies, pollack and wrasse, generally being di- or tri-chromatic), in addition to catsharks and cephalopods (being monochromatic), meaning the crab camouflage will likely be subject to selection pressure from a wide range of visual systems.

We used a computer citizen science 'game' following similar past approaches[24,25], together with protocols from other work testing search image effects and switching by observers on target detection[6,19]. The game tasked human 'predators' with finding a single crab at a time hidden against images of natural backgrounds (Fig. 2). Crab phenotypes (see Fig. 1a & Methods) were presented in runs of various lengths, allowing the participants to form search images for a given phenotype, before being switched to a new phenotype that could be of either the same or a different morph (Fig. 1b). Capture time and a range of camouflage metrics were then used to investigate search image effects. We predicted that: (i) prey camouflage strategy, and specific camouflage features (e.g. edge disruption or background colour matching), would affect capture time, with some strategies being more effective than others[26]; (ii) capture times would decrease as predators found the same phenotype repeatedly – thus building a search image – and the speed of search image formation would be dependent on camouflage strategy[19]; (iii) after forming a search image for one phenotype, the ease with which a predator would be able to switch to finding a new phenotype would vary with camouflage strategy; and (iv) switching between two phenotypes of different morphs would increase capture times more than switching between two phenotypes of the same morph. In general, we expected disruptive coloration to offer both the best protection and also to interfere most with search image formation and switching[19].

## Results

We received 1751 individual game plays, resulting in 40,354 unique crab capture events. Of these plays, 649 participants stated that they had not played the game before. There were 1669 timeout events (i.e. 3.97% of crab presentations reached timeout). Our survival data were analysed using mixed effects Cox models[27], which allow timeout data to be included in the analysis (see Methods section for full details).

**Which camouflage metrics best predict detection?** Overall, crabs took longest to find in particular when they had higher levels of disruptive coloration, see the Methods section for further details. The best overall predictors of crab capture time were: Achromatic edge disruption: This was measured using "GabRat", an edge disruption metric which compares the ratio of "true" edges (parallel to the animal's outline) to orthogonal "disruptive" edges around the entire outline, in this case using the human luminance channel (CIE L) with a sigma value of 3 ($z = -58.16$); Chromatic edge disruption: measured as above, however GabRat was performed on the human blue-yellow (CIE A) channel with a sigma value of 4 ($z = -35.07$); Pattern: The best predictor was

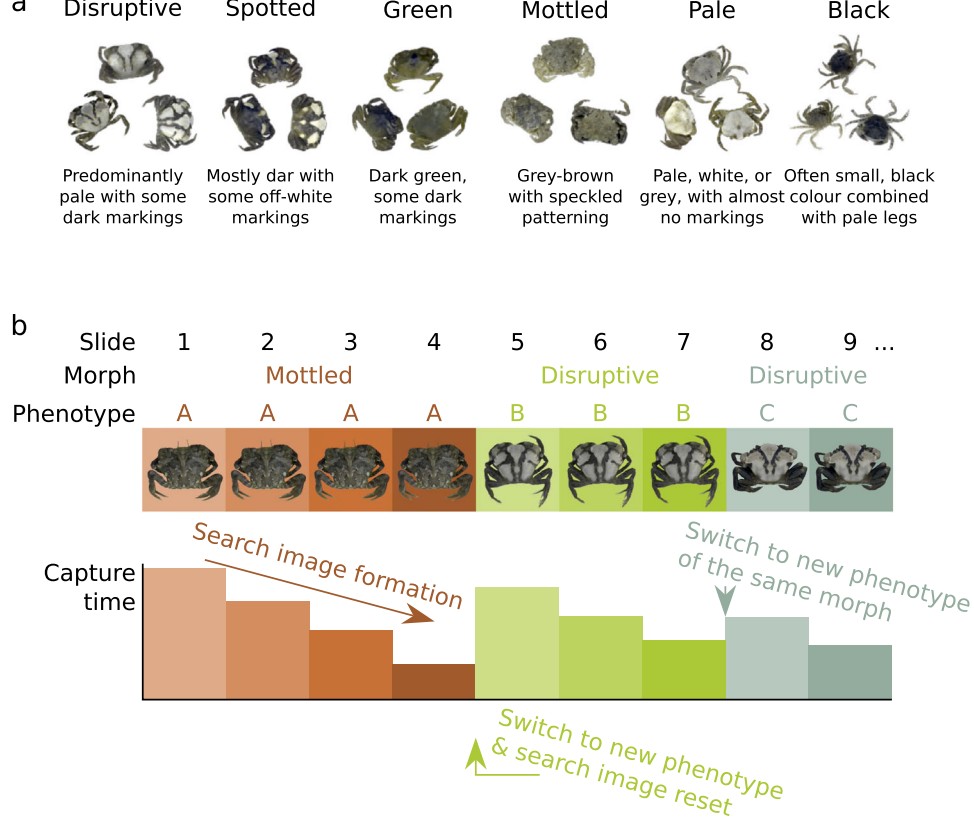

**Fig. 1 Illustration of game design. a** Crab categories illustrating the six phenotypic groups used in the search image game; **b** Illustration of the study design. This example shows the first nine slides, containing two phenotype switch events (one of which also changes the morph).

pattern energy difference between the crab and its surrounds within one body length. This metric uses bandpass granularity to characterise the pattern energy spectrum of the crab, and compares this with the background's energy spectrum ($z = 8.41$); Luminance: luminance distribution difference between the crab and its surrounds ($z = 56.20$); Colour: Euclidean distance in mean CIE AB values between the crab and its surrounds ($z = 38.36$). All $p$ values are <0.001.

**How does the number of previous encounters with the same phenotype affect capture times?** Capture times decreased as participants were shown the same crab phenotype sequentially ($z > 6.64$, $p < 0.001$ in all models), demonstrating that participants formed search images (Fig. 3). Two camouflage metrics were found to interact with the number of previous encounters with the same phenotype, these were Chromatic edge disruption ($t = -3.06$, $p = 0.002$) and Colour Match ($z = 3.34$, $p < 0.001$). Previously encountered crab phenotypes with poor colour match or low chromatic edge disruption were captured faster than novel phenotypes; moreover, if the new phenotype had a good level of colour matching or high chromatic edge disruption it was even more difficult to detect. For example, being a good colour match for novel phenotypes increased their median survival time by 42% (2442 versus 1715 ms, where colour match difference was in the 80–100th centiles), whereas for repeat-encounter phenotypes the survival advantage of a good colour match was just 33% (1842 versus 1389 ms). Likewise, having a high chromatic edge disruption for novel phenotypes increased their median survival time by 41% (2441 versus 1729 ms), whereas for repeat-encounter phenotypes the advantage of high disruption was only 32% (1829.5 versus 1386 ms). The other camouflage parameters did not show a significant interaction with the number of previous

encounters (Luminance edge disruption $z = 1.64$, $p = 0.10$; Pattern match $z = 0.82$ $p = 0.41$; Luminance match $z = 1.12$ $p = 0.26$).

**How do camouflage properties interfere with search image switching events?** This analysis considers how switching from one crab to another affects participants' capture times. More specifically, it investigates whether the difference in capture times is influenced by: (i) the difference in the level of camouflage between two successively-presented crabs; and (ii) the number of previous encounters with crabs of the same phenotype. The number of previous encounters with crabs of the same phenotype significantly influenced capture times as an interation with Luminance edge disruption ($t = -5.747$, $p < 0.001$); Chromatic edge disruption ($t = -4.360$, $p < 0.001$); Colour match ($t = 6.304$, $p < 0.001$); and Luminance match ($t = 3.569$, $p < 0.001$). There was no such interaction for Pattern match ($t = 1.464$, $p = 0.143$). See Fig. 4. The effect was most pronounced for colour matching; capturing a well-camouflaged crab of a novel phenotype (as opposed to a repeat encounter with the same well-camouflaged phenotype) took on average 60% longer (4470 versus 2800 ms). However, capturing poorly camouflaged novel phenotypes was 4% quicker than poorly camouflaged repeat encounter crabs (2384 versus 2471 ms, colour match differences in the 0–20th centiles).

**How does crab morph interfere with search image formation?** The game creates runs of phenotypes (i.e. the same crab shown a number of times, allowing a search image to build up). When switching to a new phenotype this can be of the same or a different morph as the previous phenotype, and we would broadly expect search image effects to carry over when switching to a new

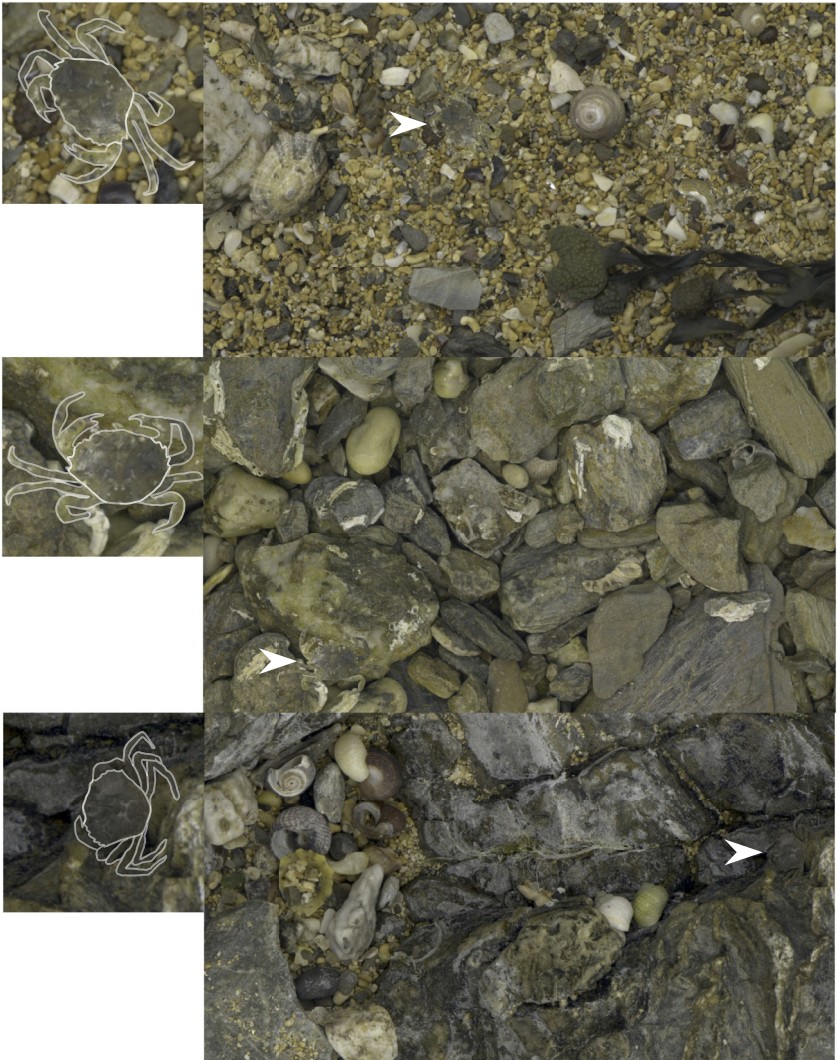

**Fig. 2 Sample game images.** Example screen shots of the crab search game. Boxes on the left show the crabs enlarged with their outline highlighted. White arrows highlight the locations of the crabs in the images.

phenotype but of the same morph. Capture time differences (the difference between capture times of the current crab versus the previous crab) were found to show an interaction between switching to a new phenotype and also switching to a new morph ($t = 4.485$, $p < 0.001$). When switching to another phenotype of the same morph capture times were increased on average by 13% (3086 versus 2723 ms). In contrast, when switching to a novel crab of a different morph (i.e. after a morph switch) average capture times increased by 28% (3396 versus 2648 ms, see Fig. 5).

## Discussion

Here, we have demonstrated that observers form search images for natural patterns found on real animals, and that search image formation varies with the level and type of camouflage. We have also shown that the ability of observers to switch between finding different individuals, and the subsequent need to re-form a new search image is influenced by the type of phenotype and camouflage properties found in both the prior and new individuals presented. In addition to illustrating these important effects of continuous variation in appearance, we also show that distinct types ('morphs') are important in affecting predator search and attention too, and that this is not simply a case of similarity of the individuals seen. Specifically, individuals of certain morph types

appear to be intrinsically more difficult to find and switch attention to when observers search for a variety of prey types.

Our results reveal a number of key factors that affect the detection of camouflaged prey, and how prey phenotypes influence predator search image formation and switching. First, we found that detection times of hidden crabs were significantly affected by the level of colour, luminance, and pattern match, as well as the degree of disruptive coloration. As expected, the level of match to the background for colour, luminance, and pattern all influenced how long participants took to find the hidden crab. Above all, in line with numerous past studies e.g.[23,26,28–31], disruptive coloration was the strongest predictor of detection times, with higher levels of edge disruption leading to longer detection times. Both background matching for colour and levels of disruption in colour (though not luminance) significantly affected the rate at which subjects improved (reduced) detection times with repeated encounters. Specifically, smaller colour differences with the background and higher levels of chromatic disruption resulted in subjects having slower capture times. This suggests that these types of camouflage are good at preventing the acquisition of information used in finding prey types, and is consistent with our earlier work using artificial target types[19,26].

The capture time differences between treatments in this study were typically in the order of hundreds of milliseconds. While this

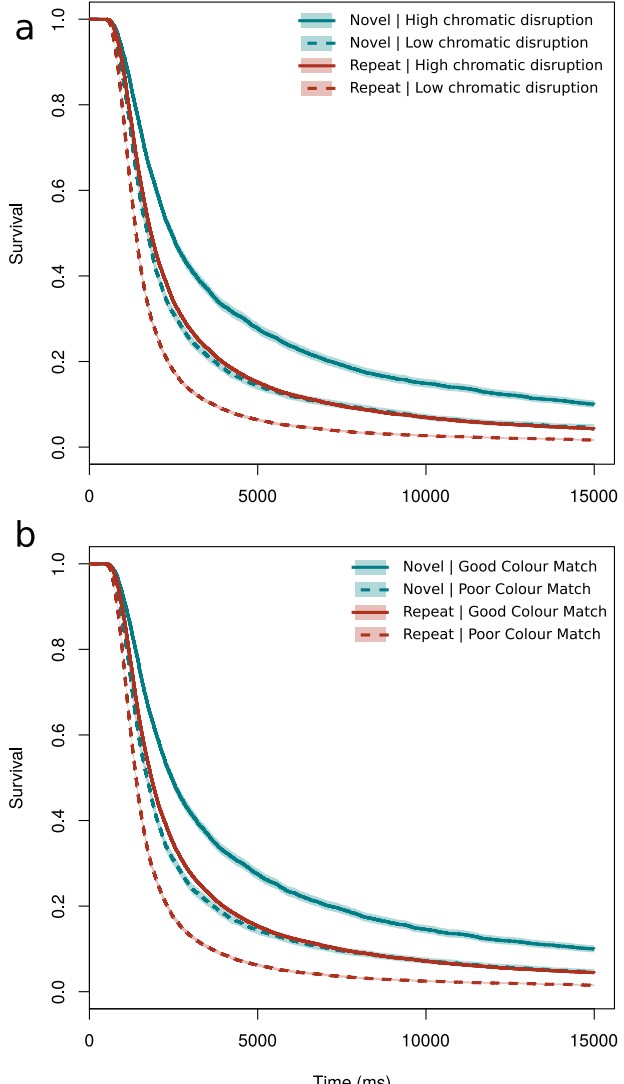

**Fig. 3 Survival analysis of novel versus repeat encounters.** Survival plots showing the interactions between novel or repeat encounters with the same phenotype, having split the data in to above- or below-median camouflage (high-low, good-poor respectively, 95% confidence intervals shown in the shaded region, based on 40,354 survival events from 1751 participants). **a** Shows chromatic edge disruption (GabRat in the CIE A channel). **b** Shows average Euclidean colour distance between the crabs and their backgrounds. These plots show a multiplicative effect where the survival advantage for novel crabs having high chromatic edge disruption, or being a good match to the surroundings is substantially higher than equivalent camouflage when participants were searching for the same crab.

may seem to be a small difference, visual fixation times in humans searching for camouflaged prey are ~150–500 ms, and fixation length (and proximity to the prey) scale with detection likelihood[32]. Therefore if the prey does not rapidly "pop out" visually on first inspection (i.e. an efficient visual search[33]), and/or an initial fixation on or near the prey fails to result in identification and capture, in a real-world situation a typical avian predator would most likely move on to new grounds without detecting the prey (unless the predator has specific reason to believe the prey is hiding in that small patch, warranting a slow "inefficient" visual search strategy); although other predatory guilds may have different hunting styles and consequently search strategies. Crucially, our experiment here is not intended to replicate a complex, three-dimensional environment with larger viewing distances and

wider area within which a predator would search, or indeed using the natural predators of these crabs. Instead, we test the potential for natural prey patterns to interfere with observer search image formation, and how this occurs. In the real-world system, the differences in detection times will almost certainly be much greater, in much the same way as if we used larger image dimensions or smaller photographs of the crabs. More important here are the statistical differences and effects we report than the absolute timings.

As predicted, when observers switched to finding a new crab phenotype it took them longer to find the new crabs when the switch occurred. This was because subjects had a search image for a different crab phenotype, and so on switching had to reform a new search image[6,19]. When switching to searching for a new crab type, capture times were affected by several metrics of appearance, and how much these differed between the previous and new crab. Particularly important aspects of camouflage were luminance disruption and chromatic disruption, and luminance matching and colour matching. Crabs with higher levels of disruption than the previously seen crabs take longer to find after switching than when switching to a new crab with similar or lower levels of disruption. Again, this is in accordance with our previous findings that higher levels of target disruption make it more difficult for observers to form search images[19]. However, here we demonstrate this effect with a much larger dataset that exposes participants to a far greater range of naturally varying phenotypes.

Our experiment also revealed interesting effects on the role of morph category in how participants detected crab types over time/experience. Specifically, when switching to a novel crab of a different morph, changes in capture times were dependent on the specific morphs involved. For example, switching from a black or green morph to a pale morph was difficult for participants, whereas switching from a mottled individual to a green crab was much easier (i.e. smaller increase in search time after switching, Fig. 5b). Put another way, in a population of crabs that are either mostly green-morphs or black-morphs, individuals would be best placed as a pale morph but at highest risk when being a spotted morph. One of the most important and surprising findings here is that the difficulty of switching between morphs is not dependent on how similar or different crabs morphs are to one another. Instead, some crab types are just much easier or harder to switch to than others, which suggests cognitive processing and receiver psychology are important for understanding effective camouflage tactics[34]. Initially, one might expect that switching from an individual of the black morph to an individual of either the green or spotted form would be relatively easier because these are all quite similar, whereas switching from black to a disruptive or pale morph should be harder since they are quite different. However, in fact, switching from disruptive to green or black is easier than switching to a pale morph. Ultimately, crabs of pale and disruptive forms are consistently harder to switch to than green or black, regardless of the first crab type seen. Spotted and mottled are more variable in how hard they are to switch to. This raises interesting questions regarding the visual features our participants were attending to; an issue poorly understood in the search image literature, and one where we might also expect interspecific differences.

In shore crabs, previous work has shown that those from habitats such as rock pools are more disruptive and more diverse in appearance among individuals than those from habitats such as mudflats, which instead rely on background matching[20,21,23]. The rock pool habitat is extremely visually diverse, and it is well known that background matching in habitats comprising diverse patch types is a challenge, while at the same time higher levels of habitat complexity relax selection for a close match[23,25,35,36]. As such, rock pool crabs rely on disruptive coloration instead, to break up the body outline[23]. Our work here also shows that the

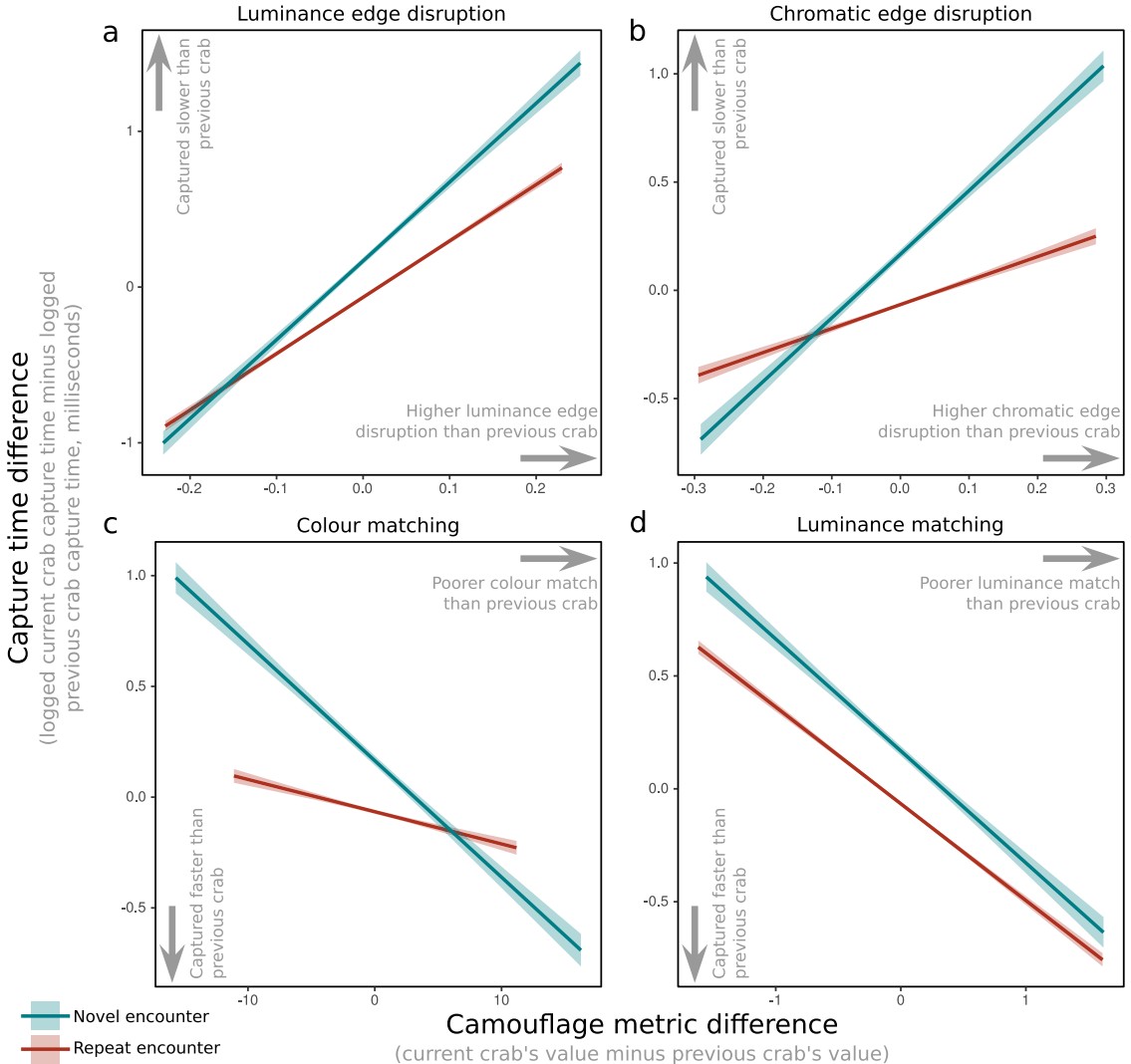

**Fig. 4 Plots showing camouflage difference versus capture time differences with each crab encounter.** Regression lines have been fitted to the raw data with standard error shown in shaded regions, based on 40,272 observations from 1751 participants. **a** and **b** (luminance edge disruption and chromatic edge disruption) show that crabs of a novel phenotype and high edge disruption take longer to be captured than crab phenotypes that have been encountered previously. **c** and **d** (colour matching and luminance matching) show that novel crabs with a better colour match to their surrounding than the previous crab take longer to be captured than crabs which have been encountered previously. This interaction is most pronounced for colour matching.

higher levels of individual variation in crabs affords them an additional benefit in inhibiting predator search image formation. The benefit really is a dual one, in that our work here and in[19] has revealed that disruptive markings are especially valuable in preventing the formation of, and switching between, predator search images. Juvenile crabs, being very small and less mobile, are to an extent 'trapped' in certain nursery habitats that can be varied in their visual composition, and as such benefit more from phenotypic variability across individuals. Larger adults – in contrast – move much more and develop a generalist background matching strategy[20,25].

What exactly it is that makes some morphs more challenging to search for and switch to, and why some forms of camouflage also impede this (e.g. disruption) needs further investigation but we suggest that this is because some phenotypes hinder the gathering of information regarding critical cues used for forming search images. That is, some pattern types may hide key features such as body shape, marking traits and so on that predators need in order to focus their attention. In addition, we have used humans here to test the above issues, but further tests are needed (albeit challenging to undertake) with natural predators.

However, humans are strongly visually guided and results from studies using humans to test camouflage concepts have been remarkably consistent with studies using non-human predators, including of search images[17,37,38]. As above, our experiment is also intentionally simplified compared to the real-world, where lighting, viewing distances, habitat area, predator species and motivation, and sources of alternative prey will all add complexity. Here, we have shown how natural prey markings can influence detection and search image formation; although it is challenging, future work should seek to incorporate some of the above aspects into wild systems to further determine the extent to which different types of animal markings, and their variability, interfere with predator search. In addition, abundance of crabs in the field is often high (especially around the middle intertidal zone) but can vary with regards to actual numbers of individuals. In some locations and patches (e.g. habitats or specific rock pools) there can be multiple crabs, others just one, or even none present. It would be valuable to explore the effect of this variation in abundance on the search image effects we report here. The strength of search images may, for example, be weakened if some patches have no crabs present at all.

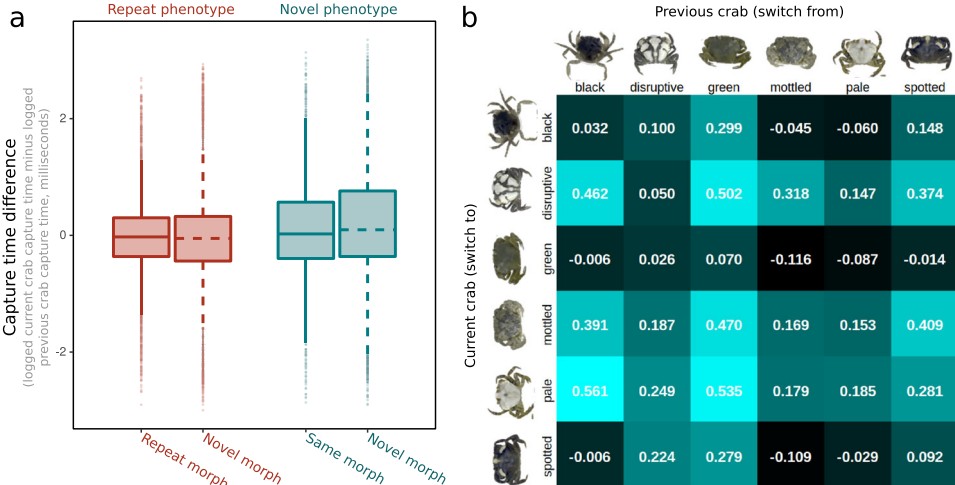

**Fig. 5 Plots showing the effects of switching between phenotypes and morphs of different types. a** Boxplots of raw data, showing the capture time interaction between repeat/novel crabs and repeat/novel morphs. When participants switched to both a new crab and a new morph, capture times were significantly increased. Boxes show interquartile ranges, points show outliers, based on 33,208 observations from 1746 participants. **b** Figure showing how each morph switch affected sequential capture times (i.e. current crab capture time minus previous crab capture time, logged ms). Positive numbers (cyan) show that the current morph is more difficult to find than the previous morph, and negative numbers (black) show that the current morph is easier to find than the previous morph.

Our findings here have a number of other important potential implications for camouflage and predator behaviour. First, given the value of disruption in impeding search images it may be that there are sometimes cases when reduced background matching may actually be optimal for reducing the likelihood of capture when predators repeatedly encounter similar prey (as is often the case in nature). Previous work[26,28] has tended to show that disruption and background matching are complementary, yet such work has focused only on how camouflage prevents initial detection. A major neglected area in research into camouflage strategies is that the appearance and camouflage of individual animals, and how successful they are, depends not only on the prey's specific appearance but also on the appearance of the wider prey population and community. Regardless, disruption is seemingly a highly successful strategy for camouflage in preventing both initial detection by naïve observers, and preventing capture when predators have repeated encounters of similar prey. Finally, our work has implications for the dynamics of polymorphisms and individual variation over time. For example, there may be trade-offs or asymmetries in the use of different camouflage appearances for preventing detection versus defeating search images. Finally, we expect apostatic selection to operate on specific prey features and types, not to simply generate or promote rare forms, such that not all morphs are equal even if their match to the background is comparable.

## Methods

**Photographs of crabs and backgrounds**. We sampled crabs and backgrounds to obtain images for the game. The population used was located in Falmouth (50.141888, −5.063811) on the south coast of the UK, comprising a stretch of shoreline encompassing neighbouring Castle and Gyllyngvase beaches. The crab habitats at the site comprise rock pools with rocky crevices with stony or gravel substrates in the pools and, lower down on the shore, increasing abundance of seaweed[21]. Together these create visually variable textures and heterogeneity in crab habitat types.

Photographs of natural backgrounds (rock pools) were taken by Samsung NX1000 digital camera converted to full spectrum and attached with a Nikon EL 80 mm lens. Background sampling was conducted along three ~100 m long transects placed parallel to the shoreline across different tide-zones (i.e. low, middle, high) spaced evenly down the beach (following[21]). Each of the backgrounds photographed were at least 5 m apart from each other (i.e. transect was subdivided approximately into 5-m-intervals) ensuring the variability in background types across transect. These sampling quadrats were photographed

during low-tide to avoid specular light reflecting back from the water. To obtain images that capture naturalistic colour variation, the images were taken in RAW format with manual white balance and a fixed aperture setting. For human visible photos as used here, we placed a UV and infra-red (IR) blocking filter in front of the lens, which transmits wavelengths only between 400–680 nm (Baader UV/IR Cut Filter). We have previously characterised the spectral sensitivity of our cameras[39]. For calibration purposes, each photograph included a grey reflectance standard, which reflects light equally at 7 and 93% between 300 and 750 nm.

Quadrats were searched for shore crabs for a period of ~5 min. We searched for crabs by raking gravel by hand, moving small boulders aside, turning seaweed over and checking crevices to ensure any crabs were unlikely to be missed. After crabs were found we transported them to laboratory facilities at the University of Exeter Penryn campus for standardised photography. During the transportation all crabs were kept on standard average grey buckets. Photographs of crabs were taken with the same camera set up as above. In the laboratory a bulb simulating D65 illuminant (Iwasaki eyeColor bulb) was used while crabs were photographed against grey standard background. We included grey standards and scale bars in the photographs. Images were then calibrated and converted to normalised reflectance images (relative to the grey standard)[39,40].

Crab images were scaled into the same pixel/mm aspect ratio to show crabs against the background images in natural size with respect to the background scale. Following past work[25], crab outlines were cut out from the image by custom software was designed (called 'autocrab') to automate the process of background subtraction. This software allowed us to step through hundreds of images, automatically loading, thresholding and flood filling background areas, saving them with an appropriate transparency channel in the correct format and resolution needed for the game. This created usable crab images for 80% of the photographs easily, with some additional cleaning up required for the rest using GIMP2 image manipulation software (https://zenodo.org/record/1101057; DOI for the source code: https://doi.org/10.5281/zenodo.1099634). The crab images were PNGs (portable network graphic) with a variable alpha level to ensure there were no jagged edges visible.

**Selection of crabs**. We aimed to ensure that we had an ecologically relevant range of crab phenotypes used in the game. We also sought to test how different types or 'morphs' of crab would affect search image formation and detection. Therefore, we used a procedure to categorise crabs into one of six categories prior the experiment. Note that, statistically crab variation may be more continuous rather than falling into true morphs, but there are a number of common crab patterns and features that frequently arise in the wild[20], potentially reflecting 'modules' of development and pattern expression. We emphasise that our aim here was not to test specifically whether shore crabs occur in discrete morphs, but rather to capture some of the variation and common features that exist in this species in order to explore the effects of different pattern types on search image formation and whether effects differ among common categories of appearance.

**Game design**. The design of the experiment generally followed the approach of previous citizen science camouflage games[24]. Ethical approval was granted by Exeter University (ID: 2015/736). Subjects were recruited via social media and word of

mouth. On loading the webpage, subjects were taken to a start screen and informed that the game was an experiment and that by playing they consented to their data being used. They were free to leave the game at any time and no personal or identifying data were collected. Subjects also asked if they had played the game before.

The game was programmed in HTML5 (including JavaScript, CSS and PHP), and was available to play on all standard internet browsers. Upon loading the game each participant was shown a series of photographs of 24 natural rock pool backgrounds (randomly sampled from 105 natural background images) with a single crab (randomly sampled from 155 natural crab images) in each image (Fig. 1). Participants were asked to detect the crab (by clicking on it) as quickly as possible, which would progress them to the next slide. If the crab was not found within 15 s the crab was highlighted with a circle for 1 s, and then the participant progressed to the next slide. During the experiment, the probability of being shown the same individual crab phenotype in the next slide was always 80% (although the crab's position and rotation, and the background image were all randomised), meaning that subjects were likely to have runs of the same individual crab in succession, often up to 10 encounters (the median run length for each crab being ~5 encounters). This approach mimicked a situation where there is no intraspecific variation in pattern, and allowed us to test which aspects of crab/morph appearance affected search image formation and switching.

**Analysis of crab appearance and camouflage.** Following our previous work testing how different types of camouflage metric predict detection[26], we analysed a large number of metrics linked to camouflage efficacy, these include edge disruption, colour, luminance (lightness), and pattern metrics. The metrics included crab-only appearance measures (such as the crab's intrinsic colour, brightness, and dominant marking size), and also comparative metrics where each crab is compared to its local surroundings (within a radius of one body-length, where body length is described as the diameter of a circle which best fits the crab's outline), and also the crab compared to the entire background image. In total there were 45 metrics, all described in Supplementary Data 1. All image analysis was performed using ImageJ v1.50[41], code available on request.

Images were converted from sRGB to CIELAB colour space before measuring them given that humans were the participants used in this study. Each crab was measured by recreating its exact position and rotation on each background for image analysis.

Luminance distribution difference was measured from the CIE L channel in 100 bins following the methods described in Troscianko et al.[26], effectively the sum of absolute differences between the crab's luminance histogram and the background or surrounding's luminance histogram. The highly variable nature of the crab's colour and background colours mean that calculating a mean colour for the background or crab may not be appropriate because it creates intermediate colours which do not represent the scene as a whole. Therefore, a colour equivalent of the luminance distribution difference method was also developed, where pixel CIE A and B values were plotted in a two-dimensional histogram to create a proportional frequency "map". Each axis had 200 bins ranging from −100 to 100, meaning the bins are smaller than the human colour discrimination threshold in CIE LAB space. The absolute differences in the crab's colour map and its background or surround colour maps were used as a non-parametric method for describing background colour matching. Edge disruption was also measured following the GabRat approach described in Troscianko et al. (2017), however in addition to measuring the CIE L image, the chromatic opponent channel images (CIE A and B images) were also measured (i.e. as a measure of chromatic edge disruption). Pattern energy difference was measured by creating a series of bandpass images, filtering each crab and surround into different spatial scales, then measuring the degree of "energy" standard deviation in pixel values) at each spatial scale to create an energy spectrum. Pattern energy difference calculates the absolute sum of energy differences at each spatial scale between the crab and its background following Troscianko et al.[26].

**Statistics and reproducibility.** Survival models were used to determine how crab capture times were affected by experimental treatments and camouflage variables. Survival models offer the ability to count crabs reaching "timeout" (where participants still could not find the crab after 15 s) as surviving up to this point (termed censored in survival models). Mixed effects survival models (coxme version 2.2–10[27]) were used to reflect the fact that within-session data are not independent. All statistical analyses were performed in R (version 3.4.4), with the raw data and R script available as supplementary material ("Supplementary Data 2", and "Supplementary Data 3" respectively). We used four different models to test each of our key predictions: (i) models ranking each of the camouflage metrics in order to find the best predictor of human performance, within each camouflage strategy the best predictor was selected and used in the subsequent tests; (ii) models testing the rate of improvement in capture time for each phenotype; (iii) models comparing the capture time and appearance of each crab relative to those of the previously encountered crab; (iv) models comparing the capture time of each crab given its morph, and the morph of the previous crab (i.e. interaction between individual phenotype and overall morph). We describe each in turn here:

First, based on our metrics of camouflage, we worked out the best predictor of human performance within each of these metrics. An example of the survival model is:

coxme(Surv(cTime, hit) ~ screenScale + playedBefore + poly (crab_circular_fit_centre_x,2) + poly(crab_circular_fit_centre_y,2) + L_GabRat_sig2.0 + crab_area + (1|sessionID), data).

This model takes into account the screen resolution, whether subjects have played before, the slide number (learning within session), the screen coordinates of the crabs (crabs in the corners of the screen take longer to find), the camouflage metric (GabRat luminance edge disruption in this example), the size of the crab (bigger crabs are easier to find), and session ID as a random factor. From these models we could calculate the metrics that were most effective in predicting detection times[26], and narrowed the metrics down to the best predictors of luminance, colour, pattern and edge disruption.

Second, we tested how the number of previous encounters with the current crab phenotype affected capture times. This is testing for speed-of-improvement within each phenotype, and how different types of camouflage (determined above) affect this. An example survival model is:

coxme(Surv(cTime, hit) ~ screenScale + playedBefore + slide + poly (crab_circular_fit_centre_x,2) + poly(crab_circular_fit_centre_y,2) + L_GabRat_sig2.0 * encounters + crab_area + (1|sessionID), data). Where 'encounters' codes for the number of previous encounters with the current phenotype.

Third, we tested capture time differences when switching between crabs, comparing the camouflage of the previous crab with the current one (note the previously encountered crab was sometimes the same phenotype, and sometimes would switch to a new one). The dependent variable (timeDiff) was log(current crab capture time) - log(previous crab capture time). The camouflage variables are calculated in the same manner, e.g. the current level of disruption minus the previous level of disruption. Here, an interaction with the number of prior encounters with the current crab phenotype shows how switching is affected by prior experience of this camouflage type. An example model is:

lmer(timeDiff ~ crab_area + pArea + playedBefore + slide + poly (crab_circular_fit_centre_x,2) + poly(crab_circular_fit_centre_y,2) + poly(pX,2) + poly(pY,2) + drpLDiff*novelCrab + (1|sessionID), diffData). The values pArea, pX and pY denote the size and screen location of the previous crab.

Finally, we analysed capture time differences when switching between each of the six crab morphs (rather than comparing camouflage metric differences), using the timeDiff value as above. An example model is:

lmer(timeDiff ~ crab_area + pArea + slide + poly (crab_circular_fit_centre_x,2) + poly(crab_circular_fit_centre_y,2) + poly(pX,2) + poly(pY,2) + slide + morphSwitch*novelCrab + (1|sessionID), morphData). Here 'morphSwitch' has two levels which describe whether a switch event was to the same, or a different morph. The random factor 'sessionID' explained almost zero variance in this dataset, and where this occurred the models were cross-validated with GLMs (see Supplementary Data 3).

**Selection of crab phenotypes.** We asked 10 naïve participants (who had no prior experience of crab phenotype discrimination) to subjectively sort images of crabs into distinct categories. People were not instructed on how many groups they should form – they were simply asked to group crabs based on their colour and patterning (i.e. phenotypic variation). This resulted in six categories (the actual numbers of the crab images representing that phenotype are given in brackets as follows): Black (22), Disruptive (15), Green (50), Mottled (28), Pale (20) and Spotted (20). Although this is subjective, we subsequently analysed the appearance of crabs from these categories and showed that 'crab morph' is a significant predictor of a range of appearance metrics, including colour, luminance, mean pattern energy, and dominant marking size ($P < 0.001$ in each case, see Supplementary Data 3 for more information). However, we emphasise that our aim here was not to test specifically whether shore crabs occur in discrete morphs, but rather to exploit the variation and common features that exist in this species in order to explore the effects of different pattern types on search image formation, and whether effects differ among common categories of appearance.

**Reporting summary.** Further information on research design is available in the Nature Research Reporting Summary linked to this article.

## Data availability
The raw data are provided as "Supplementary Data 2", and the code used to analyse the raw data are provided as an R Markdown document ("Supplementary Data 3").

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

## Acknowledgements

We thank Roger Hanlon and two anonymous reviewers for their constructive feedback, and the hundreds of people who played our online game. This work was funded by a BBSRC grant BB/L017709/1 to M.S. and J.S., and NERC Fellowship NE/P018084/1 to J.T.

## Author contributions

Study conceived by all authors (J.T., O.N., J.S. and M.S.). Crab images, phenotypic classification, and image calibration by O.N. Online game developed by J.T. following discussions with all authors. Image analysis and statistics performed by J.T. First draught written by M.S. and J.T., with all authors contributing to subsequent draughts (J.T., O.N., J.S. and M.S.).

## Competing interests

The authors declare no competing interests.
