## [Peer Review File · Communications Biology]

Reviewers' comments:

Reviewer #1 (Remarks to the Author):

This is an intriguing report on a subject of long-time interest. It is also a difficult item/subject to study experimentally. While I support this paper, I do have some difficulties with it in present form.

To begin, the title is grandiose and a bit too comprehensive - there is rather little "natural" about this study despite its importance. To wit, the study was conducted with human vision on human computer screens; there was little connection to natural light, natural animal pigments and reflectors, or the visual systems of predators. So ... the title needs to be toned down with a qualifier or two in it.

Introduction: very nice framework is presented on camouflage strategies.

Line 62: "an ideal system" I think there are other systems just as good or better ... so perhaps say "suitable" or "convenient" or even "superb?" But not ideal.

Results and Discussion:

Figure 1 is very useful. BUT ... the bottom bars in 1B got my attention quickly ... and I wanted to know the Capture time metric. When I got to Line 115 and Fig 2 I realized that human viewers were finding the crabs in 1-3 seconds ... fast!!

I think the authors fail to put in perspective what their metrics and endless statistics show: yes there are time differences in sighting crabs in different scenarios but these changes are minuscule ... for example Line 118 ... you cite a 41% increase but the actual time differences are only 1.7 vs 1.4 seconds !! Is this really a solid and biologically relevant difference when we are trying to determine eventually how a fish or diving bird predator might use or not use search image for prey detection?

I cannot grasp this as being relevant, and I study camouflage. So what about the general reader ... they are likely to question this too once they see that you report all in milliseconds.

No matter what, you have to address this in the discussion and give some rationale for assuming that such search times are relevant to how visual predators hunt. And are you assuming foveal or peripheral vision in the predator?

Figure 2 shows that ca 50-60% of crabs are detected within about 2-3 seconds - that does not impress me that (i) crabs have very good camouflage or (ii) the game with computer images etc makes it too easy for humans to find the crabs.

Line 129 "surroundings is FAR HIGHER than equivalent camouflage" is hyperbolic once the overall time framework is realized.

Figure 3. This is pretty thick going ... and it looks impressive initially, but when dealing with such short times expressed as log function the overall time differences get camouflaged (pardon the pun). Line 141: the 60% longer is only the difference between 2.8 and 4.4 seconds.

Figure 4. Nice fig, but Lines 160-162 you point out a comparison of 13% to 28% ... but the actual numbers involved are 2.7 vs 3.0 seconds! and 2.7 vs 3.4 seconds. This seems like hair splitting and raises questions about the overall framework of “visual discovery” and what it might mean in nature .. given that the eventual goal is to understand animal camouflage and how/why it evolved against animals (not humans watching computer screens).

Line 177 ... here you are a bit more careful and say “appear to be” which is appropriate.

Line 189-190 ... this seems to be a major take away message

Lines 212-214 ... another take away

Lines 231-232 ... of course another take away ... one that is becoming commonly realized in various studies

Lines 254-269. Yes these items are OK, but:

Again, you have a marvelously interesting and novel study, and have figured out one approach to a very dicey subject (quantifying search image) but your paper overinterprets the data. Moreover, you do not offer any alternative explanations of your data, you do not provide rationale for attributing such serious valuation of time changes that are only a second or three apart, nor do you point out weak parts or assumptions about the overall methodology (these would be helpful to future researchers on the subject). In summary, the way this paper is written, it comes across as “totally airtight” but it isn’t. It is a genuine step forward on a very difficult subject. My view is that if you address some of the points that struck me, you are more likely to inspire others to follow your lead as well.

All of these items can be rather easily addressed in a revision, but without these the paper falls short of its potential importance and use by the visual ecology field. You have a strong team, good crab camou model, and daring ideas that can push the field forward.

Methods: am not an expert in the detailed methods and leave that to other reviewers.

Respectfully, Roger Hanlon

Reviewer #2 (Remarks to the Author):

Using the natural variation in the colouration of shore crabs, inferred search image formation and its effectiveness are investigated. Data are collected from human observers in an on-line 'game playing' scenario. The research offers direct supporting evidence for search image formation together with its effectiveness when considered in terms of several differing colouration scenarios.

My overall impression is that the approach is interesting and thorough, and the analysis comprehensive. Although generally presented in an easy to read manner, there was, however, some lack of clarity, particularly on first reading.

In keeping with my overall impression that this is a well constructed article, I offer only two very

broad comments - the first is probably necessary, while the second is more a nice to have.

I found the results section a bit disappointing, not in terms of the breadth or specific analyses, but rather in terms of clarity. For example, it was not apparent until considerably further on, towards the end of the paper, what tests were performed and not being familiar with mixed effects Cox models, made the analysis awkward to follow. Indeed, it may simply be a problem of the ordering that affected the clarity.

The authors conclude that their findings have implications for camouflage and predator behaviour and some consideration is given to them. While I agree that they are worthy of further research, I felt that what aspects of colouration contributed to the findings, and what the effect of participants' higher perceptual/cognitive abilities might be (perhaps contrasted with other species), are important questions that deserved more, even if speculative, attention.

Reviewer #3 (Remarks to the Author):

Despite numerous studies of prey camouflage strategies, majority of research has been focused on their effectiveness to prevent initial detection, with other aspects of predator cognition being mostly neglected. By testing the effect of different types of camouflage on search image formation, this study fills this gap and brings original new findings. The experiment is well designed, the approach to data analysis appropriate, and the manuscript is concisely written. It's therefore difficult for the reviewer to detect any place for improvement. I am including only two points that could be reflected in the discussion.

(a) Is there anything known about what the natural predators of crabs are, and how their visual systems differ from that of human observers used in the experiment?

(b) In some search-image experiments, the series of pictures include also empty backgrounds without any targets (e.g. Bond & Kamil 2002). This may actually reflect the natural situation with the crabs not being present at every rock pool. Could including the empty backgrounds affect search image formation for different prey phenotypes and switching between them?

Response to reviewers

Variable camouflage patterns defeat search image formation

We thank the Editor and reviewers for their enthusiastic and constructive feedback and have endeavoured to address each of the comments, as outlined below. Please see our responses in **red text**. We have also ensured the document conforms to Comms. Biol. formatting guidelines, and as such have shortened the abstract, and moved the supplementary information section into the “Methods” section.

Reviewer #1 (Remarks to the Author):

This is an intriguing report on a subject of long-time interest. It is also a difficult item/subject to study experimentally. While I support this paper, I do have some difficulties with it in present form.

We are grateful for the reviewer’s assessment of the work’s ability to address a long-standing subject, and have now addressed their concerns about the presentation of the manuscript here, and in our revised manuscript.

To begin, the title is grandiose and a bit too comprehensive - there is rather little “natural” about this study despite its importance. To wit, the study was conducted with human vision on human computer screens; there was little connection to natural light, natural animal pigments and reflectors, or the visual systems of predators. So ... the title needs to be toned down with a qualifier or two in it.

Previous work on predator learning and search image has typically used both artificial patterns and backgrounds (with occasional exceptions), so the work here is more ‘natural’ in many regards. Nevertheless, we see the reviewer’s point so have removed ‘Natural’ and replaced it with ‘Variable’ in the title, which in some regards better explains our study anyway.

Introduction: very nice framework is presented on camouflage strategies.

Line 62: “an ideal system” I think there are other systems just as good or better ... so perhaps say “suitable” or “convenient” or even “superb?” But not ideal.

We have changed this to “excellent”.

Results and Discussion:

Figure 1 is very useful. BUT ... the bottom bars in 1B got my attention quickly ... and I wanted to know the Capture time metric. When I got to Line 115 and Fig 2 I realized that human viewers were finding the crabs in 1-3 seconds ... fast!!

I think the authors fail to put in perspective what their metrics and endless statistics show: yes there are time differences in sighting crabs in different scenarios but these changes are minuscule ... for example Line 118 ... you cite a 41% increase but the actual time differences are only 1.7 vs 1.4 seconds !! Is this really a solid and biologically relevant difference when we are trying to determine eventually how a fish or diving bird predator might use or not use search image for prey detection?

I cannot grasp this as being relevant, and I study camouflage. So what about the general reader ... they are likely to question this too once they see that you report all in milliseconds.

No matter what, you have to address this in the discussion and give some rationale for assuming

that such search times are relevant to how visual predators hunt. And are you assuming foveal or peripheral vision in the predator?

The reviewer asks (1) why we chose to report percentage rather than absolute differences, (2) to what degree the observed differences are biologically relevant, and (3) whether participants use foveal or peripheral vision. We will answer these in turn

(1) We chose to report percentage differences because these are likely more generalizable and thus informative. Absolute differences in detection time are likely influenced by a number of factors (e.g. viewing distance, lighting conditions, motivation etc). The aim of the experiment was therefore to establish whether there was a significant difference in search time rather than to establish the size of this difference. Indeed, the latter would be almost impossible to establish and would be unique to the experimental conditions used.

(2) As for the biological relevance of the absolute difference in search times. Even if we look at the absolute differences observed in the experiment (which may at first glance appear relatively small), their relevance becomes clear when considering the visual search task. Visual fixation times in humans searching for camouflaged prey are approximately 150-500ms, and fixation length (and proximity to the prey) scale with detection likelihood. Therefore if the prey does not rapidly “pop out” visually on first inspection (i.e. an efficient visual search), and/or an initial fixation on or near the prey fails to result in identification and capture, in a real-world situation the predator would most likely move on to new grounds without detecting the prey.

Moreover, our experiment was designed to take an extremely conservative measure of absolute search time. Participants had a small area to search and knew a crab to be present. Thus in natural settings where predators must search larger areas and have no prior knowledge of prey’s general locality, the search time is likely even greater.

(3) In terms of foveal versus peripheral vision, participants were free to use natural eye movements (likely to be important for assessing natural camouflage search behaviour), and eye movements combined with prey placement affect the likelihood of “efficient” versus “inefficient” search strategies, which is in turn affected by the number of distractors in the scene, and underscores the importance of using natural images for camouflage research. Ultimately the semi-random nature of eye movements, combined with random placement of the crabs and background types creates a large degree of noise in the capture times recorded, thus requiring large sample sizes

We have now clarified these points by adding the following paragraph (from line 196):

“The capture time differences between treatments in this study were typically in the order of hundreds of milliseconds. While this may seem to be a small difference, visual fixation times in humans searching for camouflaged prey are approximately 150-500ms, and fixation length (and proximity to the prey) scale with detection likelihood³⁴. Therefore if the prey does not rapidly “pop out” visually on first inspection (i.e. an efficient visual search²⁸), and/or an initial fixation on or near the prey fails to result in identification and capture, in a real-world situation the predator would most likely move on to new grounds without detecting the prey (unless the predator has specific reason to believe the prey is hiding in that small patch, warranting a slow “inefficient” visual search strategy). Crucially, our experiment here is not intended to replicate a complex, three-dimensional environment with larger viewing distances and wider area within

which a predator would search, or indeed using the natural predators of these crabs. Instead, we test the potential for natural prey patterns to interfere with observer search image formation, and how this occurs. In the real-world system, the differences in detection times will almost certainly be much greater, in much the same way as if we used larger image dimensions or smaller photographs of the crabs. More important here are the statistical differences and effects we report than the absolute timings.”

Figure 2 shows that ca 50-60% of crabs are detected within about 2-3 seconds - that does not impress me that (i) crabs have very good camouflage or (ii) the game with computer images etc makes it too easy for humans to find the crabs.

(i) The natural camouflage of *Carcinus maenas* is undoubtedly impressive (see image below), and has inspired a considerable amount of camouflage research, however see our point (ii) below for justification of these capture times.

(ii) The game difficulty is adjusted to straddle both efficient and inefficient search strategies, while also optimising the data for statistical analysis, and avoiding predator fatigue (which affects search times, and would result in people giving up prematurely). Average search times of 2-3 seconds are favoured because these will allow time for between 5-20 saccades. Note however that capture times tend to follow a log-normal distribution, and the timeout of 15 seconds (which is sufficient for an exhaustive search) should ideally not cause a significant clipping of the dataset (if too many participants reach timeout, this reduces statistical power, and interferes with predator learning). Subjectively, playing the game certainly challenges people, and we would encourage the reviewer to play themselves to assess the difficulty (crabs.sensoryecology.com). Beyond the above, consider that the subjects are sitting directly in front of a computer screen, not searching for the crabs in the wild. As we're sure the reviewer appreciates from his own work on cuttlefish, finding a hidden animal in a photo is (usually) easier than when searching in the field. In addition, here are some sample screen-shots from the game demonstrating the level of difficulty (these were the first backgrounds shown, and were not manually selected for difficulty):

Line 129 “surroundings is FAR HIGHER than equivalent camouflage” is hyperbolic once the overall time framework is realized.

We believe this language is justified given prey were roughly three times more likely to survive 5 seconds of intensive search in the higher versus lower treatments in these graphs. We have, nonetheless, changed ‘far’ to ‘substantially’ which is hopefully more appealing to the reviewer.

Figure 3. This is pretty thick going ... and it looks impressive initially, but when dealing with such short times expressed as log function the overall time differences get camouflaged (pardon the pun).

Please see above concerning the potentially critical effect a few hundred milliseconds can have and the key interpretations here.

Line 141: the 60% longer is only the difference between 2.8 and 4.4 seconds.

Indeed, which represents a large effect size (see above). In fact, a difference of 1.5 seconds or so is quite large when considered in the context of how quickly visual search over an image directly in front of a subject operates. Considered from the predator's point of view, this represents a 60% decrease in return for a given foraging bout, and from the prey's point of view is likely to increase survival chances by a similar degree.

Figure 4. Nice fig, but Lines 160-162 you point out a comparison of 13% to 28% ... but the actual numbers involved are 2.7 vs 3.0 seconds! and 2.7 vs 3.4 seconds. This seems like hair splitting and raises questions about the overall framework of "visual discovery" and what it might mean in nature .. given that the eventual goal is to understand animal camouflage and how/why it evolved against animals (not humans watching computer screens).

See above regarding small absolute detection time differences – what matters are the relative effects that reveal patterns in search behaviour.

Line 177 ... here you are a bit more careful and say "appear to be" which is appropriate.

Line 189-190 ... this seems to be a major take away message

Lines 212-214 ... another take away

Lines 231-232 ... of course another take away ... one that is becoming commonly realized in various studies

We're glad that you agree with our interpretation of our results! Given the complexities of the experimental design they are fairly complicated, so these clear takeaway messages are important. These points are all (briefly) mentioned in the abstract, so we feel they are given the prominence they are due.

Lines 254-269. Yes these items are OK, but:

Again, you have a marvelously interesting and novel study, and have figured out one approach to a very dicey subject (quantifying search image) but your paper overinterprets the data.

Moreover, you do not offer any alternative explanations of your data, you do not provide rationale for attributing such serious valuation of time changes that are only a second or three apart, nor do you point out weak parts or assumptions about the overall methodology (these would be helpful to future researchers on the subject). In summary, the way this paper is written, it comes across as "totally airtight" but it isn't. It is a genuine step forward on a very difficult subject. My view is that if you address some of the points that struck me, you are more likely to inspire others to follow your lead as well.

All of these items can be rather easily addressed in a revision, but without these the paper falls short of its potential importance and use by the visual ecology field. You have a strong team, good crab camou model, and daring ideas that can push the field forward.

Methods: am not an expert in the detailed methods and leave that to other reviewers.

Respectfully, Roger Hanlon

Many thanks for the comments above – we appreciate the perspective and agree we needed to address these issues. Please see above regarding our specific responses which

hopefully tackle them. We have also been through the paper to check for locations where we can further clarify and highlight the considerations of the paradigm and methodology here, compared to the real world. Specifically, in the Discussion we have added (from line 266):

“...our experiment is also intentionally simplified compared to the real-world, where lighting, viewing distances, habitat area, predator species and motivation, and sources of alternative prey will all add complexity. Here, we have shown how natural prey markings can influence detection and search image formation; although it is challenging, future work should seek to incorporate some of the above aspects wild systems to further determine the extent to which different types of animal markings, and their variability, interfere with predator search”

Reviewer #2 (Remarks to the Author):

Using the natural variation in the colouration of shore crabs, inferred search image formation and its effectiveness are investigated. Data are collected from human observers in an on-line 'game playing' scenario. The research offers direct supporting evidence for search image formation together with its effectiveness when considered in terms of several differing colouration scenarios.

My overall impression is that the approach is interesting and thorough, and the analysis comprehensive. Although generally presented in an easy to read manner, there was, however, some lack of clarity, particularly on first reading.

In keeping with my overall impression that this is a well constructed article, I offer only two very broad comments - the first is probably necessary, while the second is more a nice to have.

We are delighted that the reviewer found our study interesting and comprehensive, and regret any lack of clarity.

I found the results section a bit disappointing, not in terms of the breadth or specific analyses, but rather in terms of clarity. For example, it was not apparent until considerably further on, towards the end of the paper, what tests were performed and not being familiar with mixed effects Cox models, made the analysis awkward to follow. Indeed, it may simply be a problem of the ordering that affected the clarity.

We have clarified our use of cox mixed effects survival models at the start of the results section and we direct the reader to the methods section – hopefully this addresses the reviewers concerns.

The authors conclude that their findings have implications for camouflage and predator behaviour and some consideration is given to them. While I agree that they are worthy of further research, I felt that what aspects of colouration contributed to the findings, and what the effect of participants' higher perceptual/cognitive abilities might be (perhaps contrasted with other species), are important questions that deserved more, even if speculative, attention.

We welcome the invitation to speculate more on the potential differences in perceptual/cognitive abilities of humans and non-humans. Some of this ties in with the points raised by reviewer 1 (above), which we address in the additional paragraph. Additionally, we have added a section to highlight the need for replicating these effects in non-humans (from line 262):

“In addition, we have used humans here to test the above issues, but further tests are needed (albeit challenging to undertake) with natural predators. However, humans are strongly visually guided and results from studies using humans to test camouflage concepts have been remarkably consistent with studies using non-human predators, including of search images^{17,38,39}.”

See also the section above in response to reviewer 1 regarding future work building in more of the complexity of the real-world environment. However, we are hesitant to speculate too much on areas which are beyond the scope of our current study. As such we have added this line (from line 240):

“This raises interesting questions regarding the visual features our participants were attending to; an issue poorly understood in the search image literature, and one where we might also expect interspecific differences.”

Reviewer #3 (Remarks to the Author):

Despite numerous studies of prey camouflage strategies, majority of research has been focused on their effectiveness to prevent initial detection, with other aspects of predator cognition being mostly neglected. By testing the effect of different types of camouflage on search image formation, this study fills this gap and brings original new findings. The experiment is well designed, the approach to data analysis appropriate, and the manuscript is concisely written. It's therefore difficult for the reviewer to detect any place for improvement. I am including only two points that could be reflected in the discussion.

We are delighted to have such a positive assessment – many thanks!

(a) Is there anything known about what the natural predators of crabs are, and how their visual systems differ from that of human observers used in the experiment?

Indeed this is useful information, and we have added a section to the introduction to list some of the predators of green shore crabs – all visually guided, but with a range of visual systems (particularly various fish and birds, and cephalopods). The sheer number of visually guided predator species and diversity of colour vision types means that the crab camouflage is unlikely to be evolved for optimisation for a specific visual system.

(b) In some search-image experiments, the series of pictures include also empty backgrounds without any targets (e.g. Bond & Kamil 2002). This may actually reflect the natural situation with the crabs not being present at every rock pool. Could including the empty backgrounds affect search image formation for different prey phenotypes and switching between them?

This is a very good point, and something we have intended to explore in these camouflage experiments generally (but not yet done so). Indeed there are many ways the targets could be presented to predators (another example is multiple prey per screen), and this could affect learning and predator motivation. However we would argue that the *capture times* we record should scale with *detection likelihood* in real-world systems, and we hope our manuscript edits relating to this point (and reviewer 1's points above) help to justify this assumption. Assuming this holds true, then a game which uses one-prey-per-screen is far more straightforward to analyse in terms of predator learning (we would need to code for the effects of a blank screen on participant

learning, which would require larger sample sizes), and it ensures participants maintain motivation and consistent “rewards” over time. In nature, shore crab abundance will vary considerably at multiple spatial scales (e.g. patchy distributions at the scale of beaches/mudflats/rock pools/gullies). It would certainly be interesting to repeat the experiment with different probabilities of crabs present per patch/screen to explore this. We have added to the discussion (from line 272):

“In addition, abundance of crabs in the field is often high (especially around the middle intertidal zone) but can vary with regards to actual numbers of individuals. In some locations and patches (e.g. habitats or specific rock pools) there can be multiple crabs, others just one, or even none present. It would be valuable to explore the effect of this variation in abundance on the search image effects we report here. The strength of search images may, for example, be weakened if some patches have no crabs present at all”

REVIEWERS' COMMENTS:

Reviewer #1 (Remarks to the Author):

I commend the authors for a broad revision and a thorough job of it.

Overall I agree with the changes made, and as a result the ms is (to me at least) more understandable and compelling.

I do have one remaining point of challenge. The authors state "in a real-world situation the predator would most likely move on to new grounds without detecting the prey."

This is clearly untrue in my extensive diving experience. The marine fish and invertebrate predators I work with in ecosystems worldwide often have long and steady searches even in small specific areas. These can last 10s of seconds of even a minute or more in the exact same area. Anyone who has watched a barracuda on a coral reef or a slowly foraging grouper or a trumpetfish staring at one small arena for extending time would come to appreciate how patient and thorough some visual predators are in their search patterns.

I appreciate that the authors sent me a few larger images of camouflaged crabs. Indeed they are well camouflaged. But this is not clear in the actual manuscript. I suggest that the tiny images of Fig. 1C be blown up substantially ... perhaps as a separate figure rather than being crammed into 30% of Fig 1.

One other small omission:

Line 271 ... "above aspects wild systems" there is a word missing ... perhaps should read "above aspects into wild systems"^{SEP}

Nice piece of work - look forward to seeing it published soon.

Roger Hanlon

Reviewer #1 (Remarks to the Author):

I commend the authors for a broad revision and a thorough job of it. Overall I agree with the changes made, and as a result the ms is (to me at least) more understandable and compelling.

We agree the manuscript has improved and are once again thankful for your help.

I do have one remaining point of challenge. The authors state “in a real-world situation the predator would most likely move on to new grounds without detecting the prey.”

This is clearly untrue in my extensive diving experience. The marine fish and invertebrate predators I work with in ecosystems worldwide often have long and steady searches even in small specific areas. These can last 10s of seconds of even a minute or more in the exact same area. Anyone who has watched a barracuda on a coral reef or a slowly foraging grouper or a trumpetfish staring at one small arena for extending time would come to appreciate how patient and thorough some visual predators are in their search patterns.

These are indeed interesting observations! However, we believe this behaviour is better explained through two completely independent (but mutually beneficial) mechanisms: i) an ambush strategy, where the predator remains fairly still so that they can detect any motion which could reveal potential prey, and ii) use of an exhaustive (inefficient) visual search strategy. The first strategy relies on motion detection, so is quite independent from camouflage breaking. The second (inefficient search) strategy would still predict that predators should not focus their sensory systems on the same patch repeatedly because that would be even more inefficient.

Nevertheless we have adjusted the relevant sentence, and it now reads (L187):
“Therefore if the prey does not rapidly “pop out” visually on first inspection (i.e. an efficient visual search³³), and/or an initial fixation on or near the prey fails to result in identification and capture, in a real-world situation a typical avian predator would most likely move on to new grounds without detecting the prey (unless the predator has specific reason to believe the prey is hiding in that small patch, warranting a slow “inefficient” visual search strategy); although other predatory guilds may have different hunting styles and consequently search strategies.”

I appreciate that the authors sent me a few larger images of camouflaged crabs. Indeed they are well camouflaged. But this is not clear in the actual manuscript. I suggest that the tiny images of Fig. 1C be blown up substantially ... perhaps as a separate figure rather than being crammed into 30% of Fig 1.

Agreed. We have split this figure into figures 1 & 2 so that the sample images are much easier to see.

One other small omission:

Line 271 ... “above aspects wild systems” there is a word missing ... perhaps should read “above aspects into wild systems”□

Fixed – many thanks.

Nice piece of work - look forward to seeing it published soon.
Roger Hanlon